# Equity Analysis of the Green Space Allocation in China's Eight Urban Agglomerations Based on the Theil Index and GeoDetector

Xueyan Zheng [1], Minghui Zhu [1], Yan Shi [1,2,3,*], Hui Pei [1], Wenbin Nie [1], Xinge Nan [1], Xinyi Zhu [1], Guofu Yang [4] and Zhiyi Bao [1]

1  School of Landscape Architecture, Zhejiang A&F University, Hangzhou 311300, China; zhengxueyan0613@163.com (X.Z.)
2  Institute of Ecological Civilization, Zhejiang A&F University, Hangzhou 311300, China
3  Institute of Carbon Neutrality, Zhejiang A&F University, Hangzhou 311300, China
4  Artistic Design and Creation School, Zhejiang University City College, Hangzhou 310015, China
*  Correspondence: shiyan@zafu.edu.cn; Tel.: +86-136-7583-8676

**Abstract:** An urban agglomeration is a highly developed spatial area formed by integrated cities. While previous studies have analyzed green space allocation at the provincial and city scales, there is insufficient information on green space allocation in urban agglomerations. For this research, a database of green spaces in eight urban agglomerations (133 cities) in China from 2002 to 2019 was constructed to better understand the equity of green space distribution among land resources. A green space equity index (GEI) was established based on the Theil index and combined with GeoDetector to analyze the differences in urban agglomeration green spaces. The main conclusions are as follows: The sum of the GEI of China's urban agglomerations has increased significantly, rising from 3.74 in 2002 to 6.34 in 2019. The GEI value for each of the eight urban agglomerations was kept under 0.01. Polarized development has occurred within urban agglomeration cities, and the allocation of green space in megacities is relatively weak, especially in the more economically developed Yangtze River Delta and Guanzhong urban agglomerations. The average temperature, humidity, and precipitation have dominant influences in determining the GEI values. This paper provides a new perspective on the management and allocation of urban agglomeration green spaces.

**Keywords:** regional differences; green space; dynamic analysis; environmental inequalities; urbanization

## 1. Introduction

As a vital, organic part of urban ecosystems, urban green space provides a variety of ecological services to cities, including reducing urban noise [1], mitigating the urban heat island effect [2], and improving urban air quality [3]. In addition to supporting the stability of the ecosystem, green space is crucial to the well-being of residents, not only exerting a positive impact on the physical and mental health of residents but also providing them with a social place imbued with social attributes [4–7]. Against the background of new urbanization and ecological priority, reducing the imbalance in green space distribution is currently the infrastructure project that attracts the most attention from both the government and local residents.

The importance of equity research is growing worldwide, with increasing attention being given to this issue in key international documents, projects, and publications. Some of this research has mainly focused on the evolution of green space patterns and differences in green space ecological services [8]. Under the requirements of urbanization and regional economic development, scholars have shifted their research focus from a smaller range of individual parks and individual cities to a larger range of administrative regions and

even national regions. Research on the evolution of spatial heterogeneity and functional landscapes along intercity gradient urban agglomerations found that the development of urban agglomerations has led to the continuous compression and fragmentation of ecological space [9]. The level of service of green spaces in urban agglomerations has also traditionally been evaluated from the perspective of residents' needs, and the results have shown that the urban agglomerations on the eastern coast of China have a more balanced supply of green spaces and a higher level of satisfaction among residents [10]. Previous studies have lacked a holistic understanding of the green spatial distribution in the context of urbanization, especially for the determination of spatial heterogeneity, which makes it difficult to advance the equity of a green and ecological civilization.

Urban agglomerations are the highest form of spatial organization in the mature stage of urbanization development, and they are a highly efficient and typical urbanization model [11,12]. As new and important growth poles for economic development, urban agglomerations have undergone rapid urbanization and intensive land use changes, which has put enormous external pressure on the ecosystem and seriously threatened the quality of the regional habitat. The allocation of green space within urban agglomerations is directly related to the ecological environmental quality of urban agglomerations, and optimizing the allocation of green space resources across regions can assist in the improvement of the ecological service function of green space in urban agglomerations [13]. However, there is a lack of research on the spatial interaction between urbanization and green space allocation in urban agglomerations at different development levels. Therefore, there is an urgent need to focus on urban agglomerations and explore the effects of urban green space change and urbanization on the balanced distribution and quantity of green space.

For large-scale research on the quantity of green space, we used economic indicators to show the differences in the distribution of green space. Among a number of existing indices for measuring unequal distribution, the Theil index has gained broad application across different fields. The Theil index is a tool for measuring the degree of inequality in the allocation of green spaces across different urban agglomerations. The Theil index goes beyond the traditional quantitative comparison of green space data and is applied at two scales: the region and its internal component units. The higher the Theil index value is, the greater the inequality in green space allocation. The Theil index provides a clear view of the share of fluctuations in the data of the internal component units of the regional changes. This information can be used to develop more equitable policies for green space allocation and ensure that all residents have access to adequate green spaces.

In this study, by using both the Theil index and GeoDetector, the extent of inequality in green space allocation within urban agglomerations as well as the factors contributing to this inequality are identified. The objectives of the study are as follows: (1) To indicate the dynamics of green space equity patterns at the urban agglomeration level and at the intraurban agglomeration level. (2) To analyze the relation using socioeconomic and geographical variables of green space allocation at the urban agglomeration level. This study can provide a comparative baseline for the improvement of green space allocation in urban agglomerations, which is crucial for improving the ecological environment of urban agglomerations and promoting their sustainable development.

## 2. Materials and Methods

### 2.1. Selection of Research Scope

The urban agglomerations selected are sufficiently representative; those with the best economic development are located in different climatic conditions and geographical locations. This study selected eight Chinese urban agglomerations containing 133 cities: the Yangtze River Delta urban agglomeration (Y-R-D), the Pearl River Delta urban agglomeration (P-R-D), the Beijing–Tianjin–Hebei urban agglomeration (B-T-H), the Middle Reaches of the Yangtze River urban agglomeration (M-R-Y-R), the Chengdu–Chongqing (Yu) urban agglomeration (C-Y), the Guan Zhong urban agglomeration (G-Z), and the Central Plains urban agglomeration (C-P). One local-level urban agglomeration, the Heilongjiang–Jilin

urban agglomeration (H-J) was also selected. The urban agglomeration levels and socioeconomic data are shown in detail in Table 1. By the end of 2019, the built-up area covered by this study accounted for 45.6% of the country, and the population reached 65.9% of the national urban population.

**Table 1.** Detailed information on urban agglomeration development.

| Name of Urban Agglomeration | Level | Date Established | Number of Cities | Development Goals | Urban Population (Million) | Built-Up Area (Ha) | GDP Per Capita (RMB) |
|---|---|---|---|---|---|---|---|
| Yangtze River Delta urban agglomeration | National | 5/2010 | 20 | Economic ☑ Ecological ☑ Agriculture ☒ | 6633 | 461,831 | 55,598 |
| Beijing–Tianjin–Hebei urban agglomeration | National | 4/2015 | 10 | Economic ☑ Ecological ☑ Agriculture ☒ | 4305 | 385,714 | 51,902 |
| Pearl River Delta urban agglomeration | National | 9/2015 | 9 | Economic ☑ Ecological ☒ Agriculture ☒ | 4705 | 451,111 | 48,118 |
| Middle Reach of Yangtze River urban agglomeration | National | 3/2015 | 31 | Economic ☑ Ecological ☒ Agriculture ☒ | 3553 | 405,370 | 37,996 |
| Central Plains urban agglomeration | National | 12/2016 | 24 | Economic ☑ Ecological ☑ Agriculture ☑ | 2703 | 341,302 | 45,111 |
| Guanzon Plain urban agglomeration | National | 1/2018 | 13 | Economic ☑ Ecological ☒ Agriculture ☒ | 1558 | 126,203 | 33,895 |
| Chongqing–Chengdu urban agglomeration | National | 11/2018 | 16 | Economic ☑ Ecological ☑ Agriculture ☑ | 3624 | 391,806 | 37,046 |
| Heilongjiang–Jilin urban agglomeration | Local | 9/2016 | 10 | Economic ☑ Ecological ☒ Agriculture ☒ | 1588 | 184,990 | 34,340 |

Statistics on population, built-up area, and GDP per capita are all for 2019.

Some of these urban agglomerations are gradually developing into economic centers of the country [14]. The Y-R-D, P-R-D, and B-T-H are the three pillars of economic growth in China [15]. In 2018, the Y-R-D, P-R-D, and B-T-H accounted for 24.29% of China's total population while representing only 5.18% of the country's land area; these urban agglomerations accounted for 38.53% of China's total economic output [12]. Moreover, the per capita disposable incomes in these three urban agglomerations are the three highest in the country. New economic agglomerations are also emerging, such as the M-R-Y-R. This region has been undergoing rapid urbanization and industrialization [16]. The C-Y is the only national urban agglomeration in China that accounts for both economic development and modern agricultural activities. Other nascent urban agglomerations are also gradually taking shape, such as the G-Z, the C-P, and the H-J. However, due to their weak foundation, late start, and unclear development goals, the development speed of these agglomerations is relatively slow.

The research object of this paper includes various geographical locations (Figure 1). Geographically, the B-T-H, Y-R-D, and P-R-D are located in the developed regions of northern, eastern, and southern China, respectively [12,17]. The Y-R-D is the only world-class urban agglomeration in China. Located in the Yangtze River basin and spanning three provinces [12], the Y-R-D has the largest built-up area in China. The B-T-H includes the capital of China, Beijing, and nine other cities. The P-R-D is located on the southern coast of China and is the second-largest established urban agglomeration in China. The M-R-Y-R is one of the most important urban agglomerations in China's inland basin. It covers three provinces, with 31 cities in total [18], and has the largest volume among China's urban agglomerations. The C-Y is located in the Sichuan Basin and contains two cities, Chengdu and Chongqing, as its center, as well as 14 other cities in total. It has experienced significant population growth, growing by almost 190% over the last 20 years [19]. The C-P is a heavily

resource-constrained urban agglomeration in western China. It consists of the provinces of Henan, Hebei, Anhui, and Shandong [18]. However, the population concentration caused by its rapid urbanization continues to put pressure on the ecological environment in this area [20]. The G-Z urban agglomeration is located in the center of China's inland and is a crucial fulcrum of the Asia–Europe Continental Bridge and an essential gateway between the western and the eastern regions. The H-J urban agglomeration has a deep industrial base and is a valuable part of China's heavy industry development.

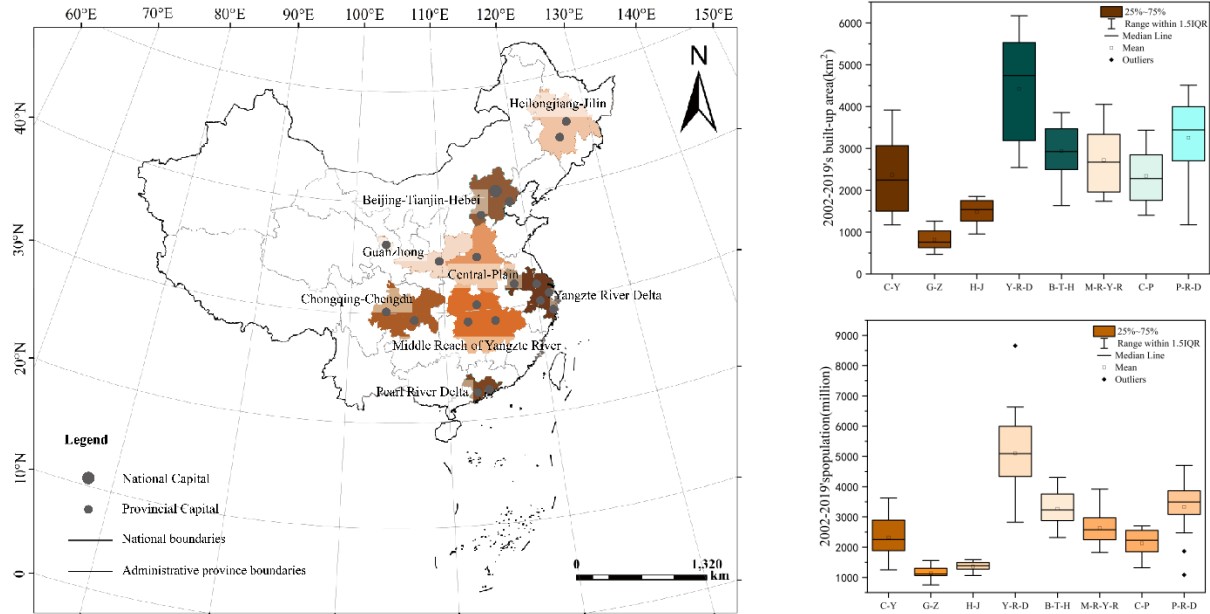

**Figure 1.** Locations, built-up areas, and population sizes of eight urban agglomerations.

## 2.2. Data Collection

In this study, according to the National Bureau of Statistics of China and previous studies, the built-up area refers to the area containing well-developed urban construction, which usually includes the urban area and part of the urban-rural interface [21,22]. The city population is defined as the population permanently living in the focal city. Urban green spaces (UGSs) are defined as areas within the built-up areas of cities containing either all natural, artificial, or semiartificial forms of vegetation. Per capita disposable income is defined as the sum of final consumption expenditure and other nonobligatory expenditures and the per capita savings available to households. Per capita GDP is defined as the ratio of the gross product produced in a region in a year to the population of that region and is used as a measure of the standard of living of residents in the region.

The data used in this study to measure the socioeconomic conditions of urban agglomerations are their population size, disposable income per capita, and GDP per capita. The socioeconomic data and natural data (Table 2) used to verify the spatial distribution characteristics in this study were obtained from the Resource Environment Science and Data Center https://www.resdc.cn/ (10 January 2023). The urban agglomeration GDP data, Chinese population density data, average temperature, average humidity, and average precipitation data of the urban agglomerations in 2019 are raster data with a resolution of 1 km ∗ 1 km. The SRTMDEM digital elevation data of Chinese urban agglomerations were obtained from the geospatial data cloud website http://www.gscloud.cn (11 January 2023). Using ArcGIS 10.3 software, the urban agglomeration elevation DEM data were surface-analyzed to calculate both the slope and slope direction.

**Table 2.** Variables and statistical descriptions.

| Variables | Definition | Type | Mean | S.D. | SUM | Min. | Max. |
|---|---|---|---|---|---|---|---|
| | **Dependent variables** | | | | | | |
| **GEI (i)** | Theil index of the number of green spaces in urban agglomerations | | 0.01141 | 0.01265 | 1.64325 | 0.000171 | 0.10412 |
| | **Independent variables** | | | | | | |
| **UP** | Total resident and transient population in the built-up area (10,000) | Socioeconomic | 2660.32 | 1392.89 | 383,086.16 | 752.66 | 8657.80 |
| **BUA** | Built-up area (ha) | | 254,446.35 | 127,519.23 | 36,640,300 | 46,948 | 617,035 |
| **PCDI** | Disposable income per capita (CNY) | | 22,084.70 | 11,679.63 | 3,180,195.80 | 6238.88 | 55,598.43 |
| **A-ATEMP** | Average annual temperature (°C) | | 15.23 | 4.67 | 2193.65 | 4.85 | 23.20 |
| **A-ARH** | Annual average humidity (%) | Natural | 66.25 | 9.27 | 9539.65 | 52 | 82 |
| **A-AP** | Average annual precipitation (mm) | | 993.49 | 556.16 | 140,082.35 | 172 | 2939.70 |

## 2.3. Analysis Methods

A detailed description of each step is introduced in this section. Figure 2 presents the workflow for this study.

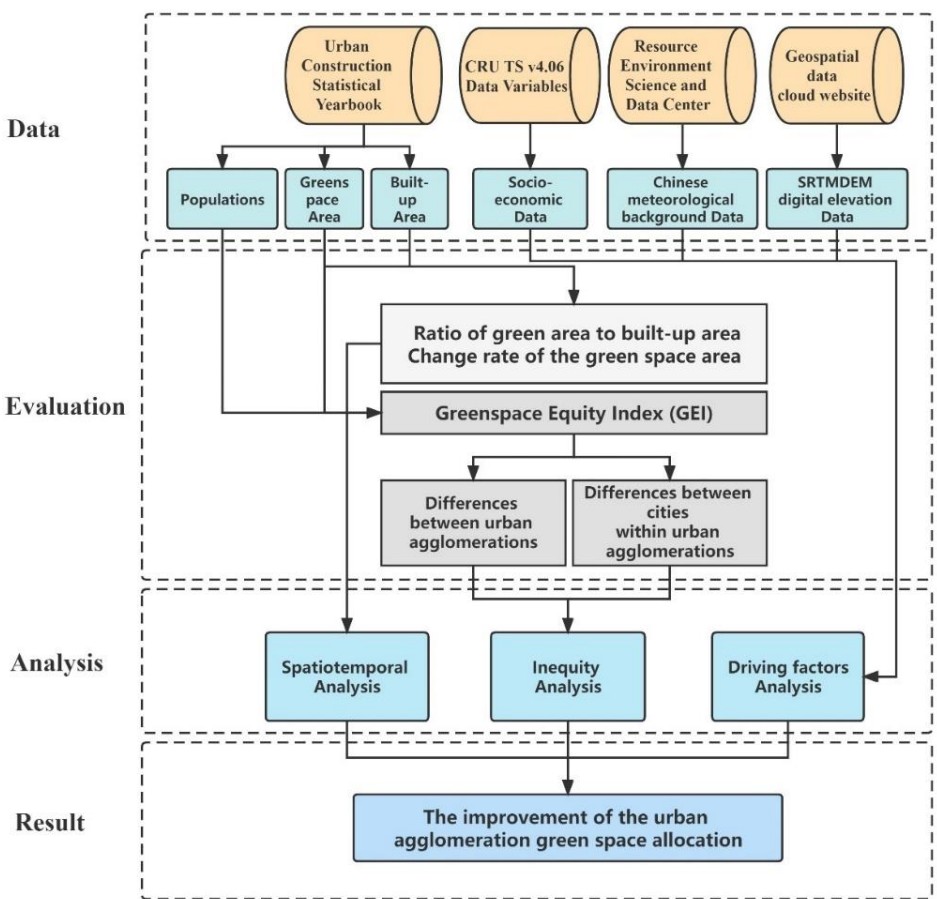

**Figure 2.** Workflow chart for this study.

Step 1. The green space equity index (GEI) and the dynamic changes in the green area in urban agglomerations were calculated using the values for the green area and population. Step 2. We analyzed the results of green space distribution equity in the urban agglomerations based on the results of the GEI index. Step 3. GeoDetector was used to analyze the spatial correlation, and Spearman's correlation was used to analyze other driving factors. Step 4. Ways to optimize the green space distribution in urban agglomerations are suggested based on the above results.

2.3.1. Theil Index

The Theil index [23] measures the degree of regional variation and decomposes the overall variation into internal and external variations at different scales [24]. The Theil index uses the concept of entropy in information theory to calculate the degree of inequality. The rationale for preferring Theil's T statistic is that, compared to other inequality indices, the Theil index usually has less stringent data requirements, which is a benefit when group data are more readily available than individual survey data. In addition, one of the most important features of Theil's index is its decomposability, as it can be decomposed into the sum of the "between-group" component GEIi and the "within-group" component GEIj (Figure 3). This study accounted for the differences in the number of urban agglomerations in China and the complexity of city counting. To summarize, we assigned demographic significance to the Theil index, as weighted by demographics, to give us a better understanding of how the differences in green space distribution truly look in the context of population surges. Notably, to eliminate the error caused by the number of cities, we adjusted part of the Theil index decomposition, using the median value to eliminate the number error.

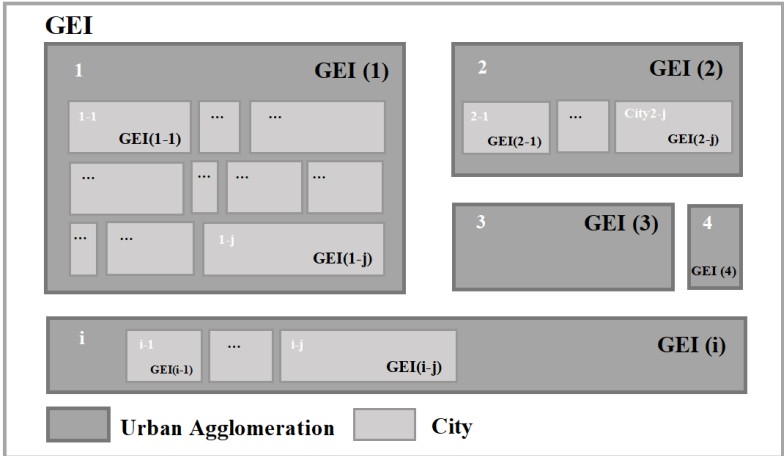

**Figure 3.** Schematic diagram of the composition of GEI indicators.

The equilibrium index of the built-up area green equality index (GEI) was established to quantify the unevenness and dynamic changes among green areas in urban agglomerations, using the Theil index as the base model.

According to the Theil index:

$$\sum_{i=1}^{Ni} \left(\frac{xi}{X}\right) \times ln\left\{ \left(\frac{xi}{X}\right) / \left(\frac{yi}{Y}\right) \right\} \tag{1}$$

Then, the overall difference is:

$$GEI = Ni - [GEI\ (i) + GEI\ (j)]$$

The between-group component is:

$$GEI\ (i) = \Sigma\ (Xi/X) \times ln\ [(Xi/X)/(i/Y)] \tag{2}$$

The within-group component is:

$$GEI\ (j) = \Sigma\ (xi/Xi) \times ln\ [(xij/Xi)/(yij/Yi)] \tag{3}$$

where i = 1, 2, 3, 4, 5, 6, 7, 8, Ni is the sum of the maximum difference values of the i urban agglomerations, GEIi is the difference between urban agglomerations, and GEIj is the difference within urban agglomerations. Y and X represent the urban agglomerations'

green space (UGS) and the total resident and transient population in the built-up area (UP), respectively, Y I and X I represent the UGSA and UP of the ith urban agglomeration, respectively, yij represents the BUGS of the jth city in the ith urban agglomeration, and xij represents the UP of the jth city in the ith urban agglomeration. The value of 0, assigned when the BUGS ratio and UP ratio of each city group are the same, indicates absolute equilibrium. The larger the value is, the larger the difference.

The calculation of the Theil index requires two data sets, each of which can be classified into mutually exclusive and completely exhaustive groupings. This requirement makes annual state statistical yearbooks an attractive data source [25].

### 2.3.2. GeoDetector Analysis Method

GeoDetector is a set of statistical methods for detecting spatial differentiation and revealing the driving forces behind it. This tool can analyze the relationship between different factors and the distribution of green spaces to determine which factors exert the greatest impact on the distribution of green spaces in a region. GeoDetector consists of four detectors. Among them, the statistical principles of the divergence and factor detectors are detecting the spatial divergence of Y and detecting how much of the spatial divergence of attribute Y is explained by a certain factor X, respectively. The q-value metric is also used. The general calculation formula is as follows:

$$q = 1 - \frac{\sum_{h=1}^{L} N_h \sigma_h^2}{N \sigma^2} = 1 - \frac{SSW}{SST}$$

$$SSW = \sum_{h=1}^{L} N_h \sigma_h^2, SST = N \sigma^2 \tag{4}$$

where $h = 1, ..., L$ is the strata of variable Y or factor X, i.e., the classification or partition, $N_h$ and $N$ are the number of cells in the strata and the full area, respectively, and $\sigma^2 h$ and $\sigma^2$ are the variances in the Y values in the stratum h and the full area, respectively. *SSW* and *SST* are the within sum of squares and total sum of squares, respectively. The value domain of $q$ is [0, 1]. A larger value indicates that the spatial heterogeneity of Y is more pronounced; when the stratification is generated by the independent variable X, a larger q value indicates that the independent variable X has a stronger explanatory power regarding the attribute Y and vice versa. In the extreme case, a q value of 1 indicates that factor X completely controls the spatial distribution of Y. A q value of 0 indicates that factor X has no relationship with Y. A q value indicates that X explains $100 \times q\%$ of Y.

### 2.3.3. Data Analysis

According to the basic principle of fitting, these data need to be processed normally [21] (in transformation), and then correlation analysis and regression analysis must be conducted. Due to the large numerical gap among different variables, using Pearson correlation analysis may affect the accuracy of the results, so Spearman correlation analysis was selected instead. Multiple linear regression analyses can be used when most of the variables are significantly correlated. A two-tailed significance test was used, with one dependent variable GEI (i), three socioeconomic factors, and three natural factors based on using the values of the hydrothermal gradient as independent variables.

## 3. Results

### 3.1. The Change Trends of Green Space Area in Urban Agglomerations

The improvement in green space area is obvious. From 2002–2019, the UGS area in Chinese urban agglomerations increased by 348% (Figure 4b). Among them, the Y-R-D had the most significant increase in UGS, reaching 520%; the B-T-H increased by only 156%. Moreover, as the three most economically developed urban agglomerations in China, the green space areas of the Y-R-D, B-T-H, and P-R-D were consistently maintained at average levels between 2002 and 2019. In contrast, urban agglomerations with poorer natural

foundations, such as the H-J and G-Z, have less green space. This can be partly attributed to the economic push for green space growth.

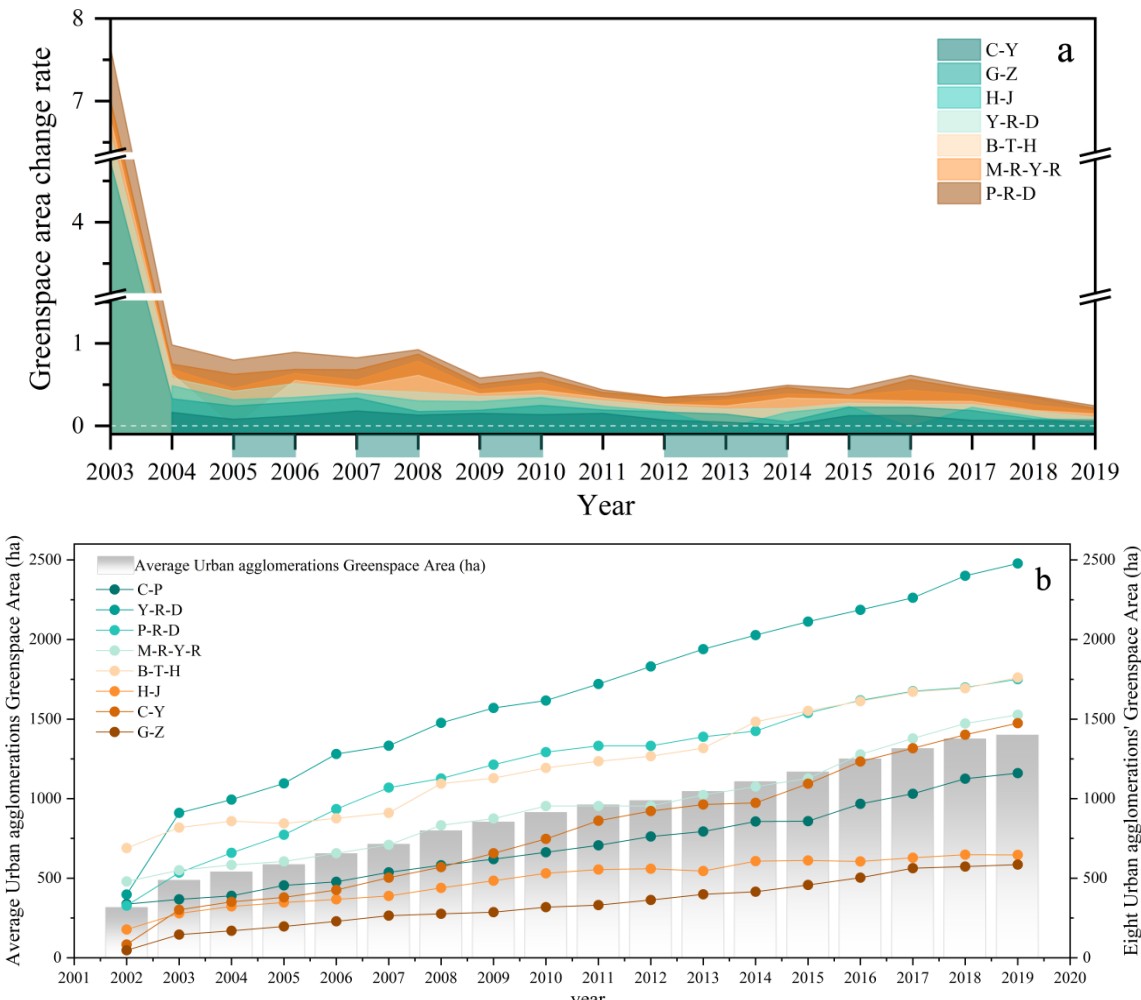

**Figure 4.** The annual average growth rate of green space and panel data of green space area across 8 urban agglomerations in China. (**a**) The change rate of the green space area. The dotted line graphs rendered in different colors in (**b**) represent the changes in the UGS of urban agglomerations, and the gray histogram shows the average level of UGS in these urban agglomerations.

The growth rate of green space in urban agglomerations has gradually slowed over time (Table 3). The magnitude of the change in the ratio of green area to built-up area has gradually stabilized. The ratio increased more significantly in the first two years, showing an increase of 18% in the Y-R-D. In the last decade, the ratio of UGS to built-up area did not increase by more than 4% in any of the eight urban agglomerations. The panel data for green space area contain similar findings (Figure 4a). The UGS in China's urban agglomerations experienced significant growth in 2003–2004. However, after 2008, the growth rate of UGSs in urban agglomerations began to gradually decline. This slowdown not only represents a slowdown in speed but also indicates negative growth. The H-J lost UGS during 2012–2013 and 2015–2016; the B-T-H lost 2% of its total UGS in 2004–2005. In contrast, other urban agglomerations experienced a slower greenfield growth trend for the first time in 2011 and again in 2017. From this, we can infer that the UGS of the urban agglomeration has almost reached its saturation point. While it is true that the number of UGSs in China's urban agglomerations has increased significantly over the past 20 years, the increase in the number of UGSs does not fully represent a decrease in green space disparity.

**Table 3.** Changes in the ratio of green area to built-up area and population size changes at nine-year intervals across eight urban agglomerations in China over the past 18 years. (−) and (+) represent the increase and decrease in changes in UGS/built-up area over three adjacent years, respectively.

| Urban Agglomeration | Changes in UGS/Built-Up Area (%) | | | | | | | | Population (Million) | | |
|---|---|---|---|---|---|---|---|---|---|---|---|
| | 2002–2004 | 2004–2006 | 2006–2008 | 2008–2010 | 2010–2012 | 2012–2014 | 2014–2016 | 2016–2019 | 2002 | 2010 | 2019 |
| Y-R-D | 0.18 (+) | 0.03 (+) | 0.04 (+) | 0.03 (+) | 0.01 (+) | 0.03 (−) | 0.00 (+) | 0.01 (+) | 12.50 | 22.23 | 36.24 |
| B-T-H | 0.16 (+) | 0.00 (−) | 0.04 (+) | 0.00(+) | 0.01 (+) | 0.01 (+) | 0.01 (+) | 0.04 (−) | 7.53 | 10.82 | 15.58 |
| P-R-D | 0.11 (+) | 0.01 (−) | 0.04 (+) | 0.02(+) | 0.00 (−) | 0.01 (+) | 0.02 (+) | 0.02 (+) | 10.64 | 13.60 | 15.88 |
| M-R-Y-R | 0.17 (+) | 0.04 (+) | 0.00 (−) | 0.01(+) | 0.00 (+) | 0.00 (−) | 0.00 (+) | 0.00 (+) | 28.26 | 49.97 | 66.33 |
| C-P | 0.03 (+) | 0.02 (−) | 0.05 (+) | 0.02(+) | 0.01 (+) | 0.03 (+) | 0.01 (+) | 0.01 (+) | 23.19 | 31.95 | 43.05 |
| G-Z | 0.03 (+) | 0.03 (+) | 0.02 (+) | 0.02(+) | 0.04 (−) | 0.01 (+) | 0.01 (+) | 0.01 (+) | 18.51 | 25.20 | 35.53 |
| C-P | 0.01 (+) | 0.02 (+) | 0.02 (+) | 0.00(+) | 0.01 (+) | 0.01 (+) | 0.02 (+) | 0.02 (+) | 13.88 | 22.21 | 27.03 |
| H-J | 0.00 (+) | 0.06 (+) | 0.03 (+) | 0.01(+) | 0.01 (−) | 0.01 (+) | 0.00 (+) | 0.01 (−) | 10.84 | 31.74 | 47.05 |

### 3.2. GEI Dynamics in Urban Agglomerations

In the past 20 years, UGS equity in China's urban agglomerations has gradually increased, rising from 3.74 to 6.34 (Figure 5). The results show that the difference in UGS among urban agglomerations was the most significant in 2002. Between 2002 and 2004, the GEI grew more than 1.5 times. Subsequently, it reached the first highest value between 2008 and 2009 and continued to change slightly in the following seven years. In 2017, the GEI value decreased significantly, and then it returned to the original maximal level in 2018.

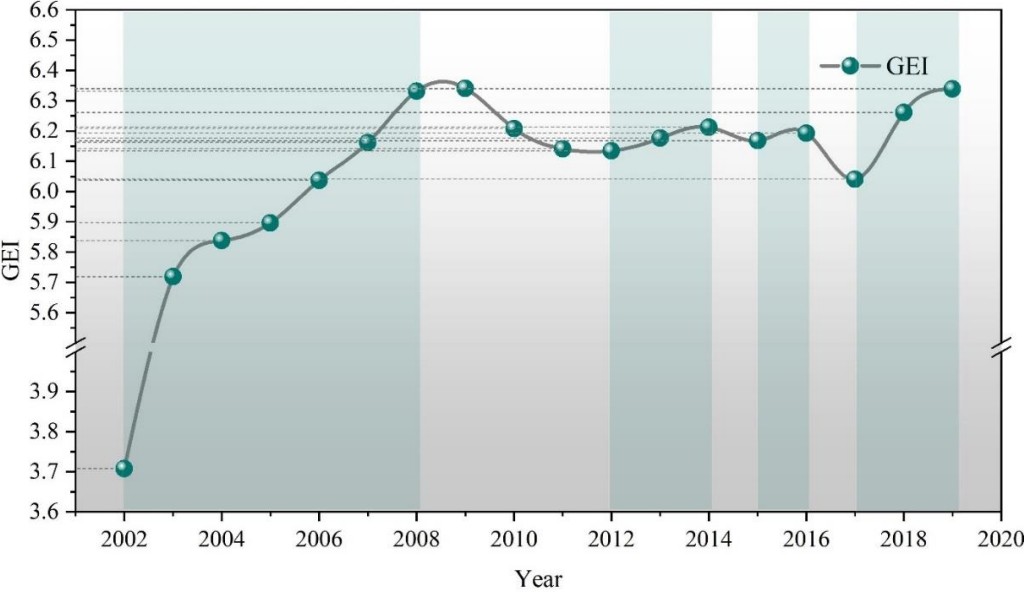

**Figure 5.** GEI results of urban agglomerations in China. The green area indicates the year in which the GEI value increased.

### 3.3. The GEI Results between Eight Urban Agglomerations in China

The GEIs of eight urban agglomerations were analyzed (Figure 6c,d), and the results revealed an unbalanced spatial distribution of UGS. In particular, 2009–2010 appears to be a critical period for the reduction in greening area disparities. The research time period was divided into two stages: the first stage, ranging from 2002–2009, was "the period of rapid reduction in differences"; the second stage, ranging from 2010–2019, was "the period of stable differences". In 2002, the differences in UGSs in China's inland urban agglomerations were not significant. In contrast, the UGSs of economically developed urban agglomerations, such as the Y-R-D and B-T-H, were quite different. These situations changed in 2019, and the difference in UGSs between the two central urban agglomerations (the C-P and G-Z) gradually surpassed that of other urban agglomerations.

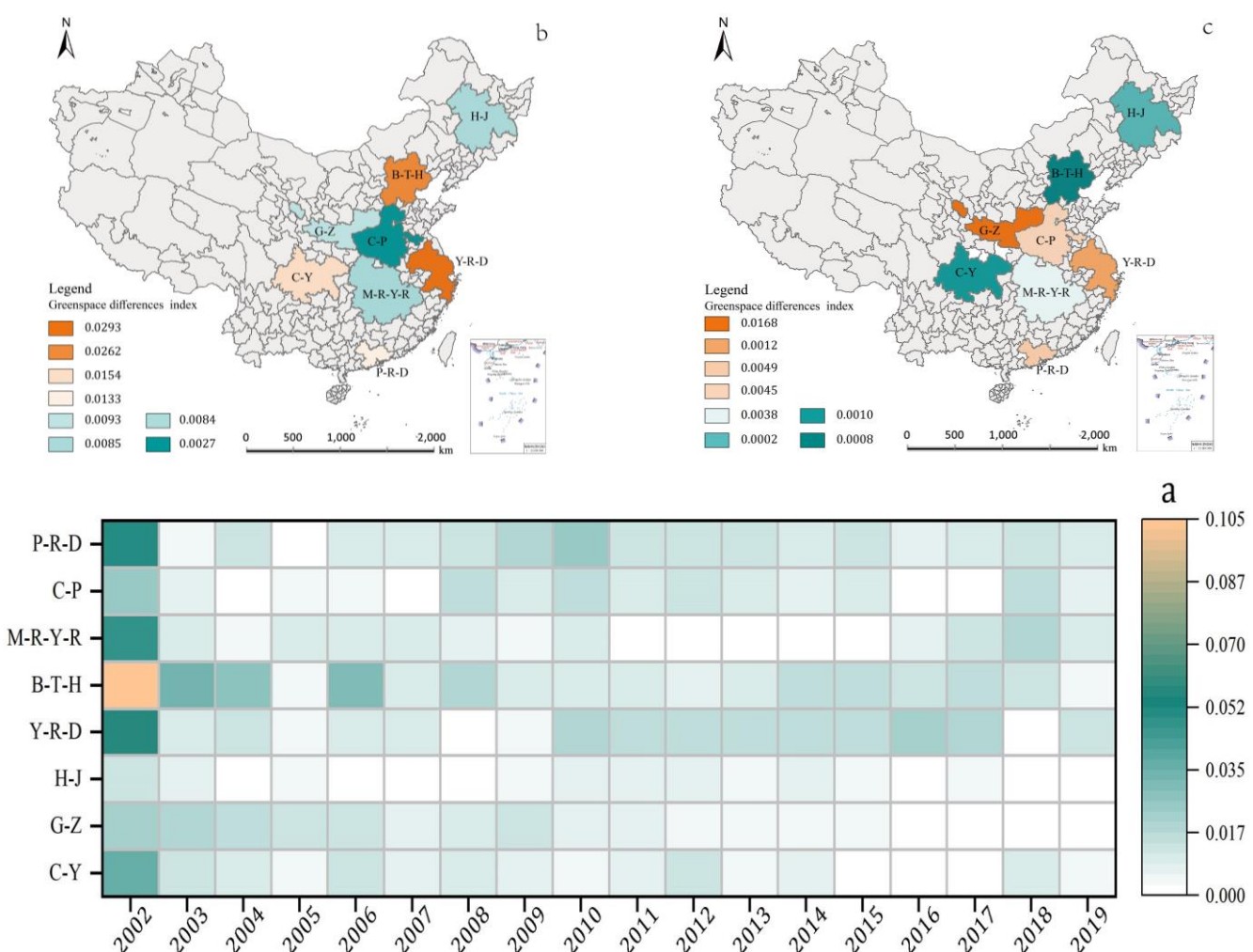

**Figure 6.** (**a**) The GEI values of eight urban agglomerations from 2002 to 2019; (**b**) the GEI value of urban agglomeration in 2002; (**c**) the GEI value of urban agglomeration in 2019. To eliminate the statistical error caused by the varying number of cities contained in different urban agglomerations, we optimized the model indicators and used the median as a parameter in the GEI model.

The GEI results of some urban agglomerations are very noteworthy. The GEI of the Y-R-D maintained an extremely high value during the study period, showing a fluctuating state. As a developed region with a fine natural base and suitable climatic conditions, it does not seem to be as green as it could be, and panel data show that it has the highest UGS but that its GEI is consistently extremely high in weighted conditions. The B-T-H had the highest GEI in 2002, with the degree of GEI decreasing over time, but the trend has been more volatile; similarly, the degree of GEI of the M-R-Y-R has weakened. The P-R-D had a high GEI in 2002, but the degree of GEI has weakened, and the trend has been more volatile. On the other hand, the UGS differences among the H-J, G-Z, and C-Y are clearly not in a healthy state. Economic development has a strong attraction for the population. With the development of urbanization, an increasing number of people gather in economically developed cities, such as those in coastal areas, and the population flows across the underdeveloped areas in the central part [26]. The urbanization process has slowed down, and thus, green space has been preserved.

### 3.4. GEI Results for the Cities within Urban Agglomerations

We estimated the degrees of GEI in 133 prefecture-level cities in China from 2002 to 2019 (Figure 7). Among them, there are three world-class metropolises (Shanghai, Beijing, and Guangzhou), 16 large cities (Shenzhen, Tianjin, Shijiazhuang, Nanjing, Suzhou,

Chongqing, Chengdu, Changchun, Harbin, Xian, Lanzhou, Zhengzhou, Wuhan, Changsha, Nanchang, and Hangzhou), and 114 other cities. As illustrated in Figure 6b, in 2002 at the individual city level, there was a very significant polarization of UGS differences among cities. It is particularly important to note that 46 cities underwent an increase in UGS differences compared to 2002. Among them, 13 cities belong to the C-P, 13 to the M-R-Y-R, 10 to the Y-R-D, 4 to the P-R-D, 2 each to the H-J and C-Y, and 1 each to the B-T-H and G-Z. It is impossible to ignore that many core cities of urban agglomerations are experiencing negative growth in UGS distribution differences, including mega-cities, such as Shanghai and Tianjin, and economic centers of urban agglomerations, such as Changsha and Zhengzhou. From the perspective of urban agglomerations, the GEI gap between core and peripheral cities is also very significant, and the ecological environment of core cities is being tested. Moreover, within the same urban agglomeration, there is a very large gap in UGS differences between those metropolitan cities and municipalities directly under the central government and small and medium-sized cities (representing the rest of the cities in the agglomeration). As illustrated in Figure 6c, the polarization of UGS differences in Chinese cities, in terms of both individual cities and urban agglomerations, was mitigated in 2019 compared to 2002.

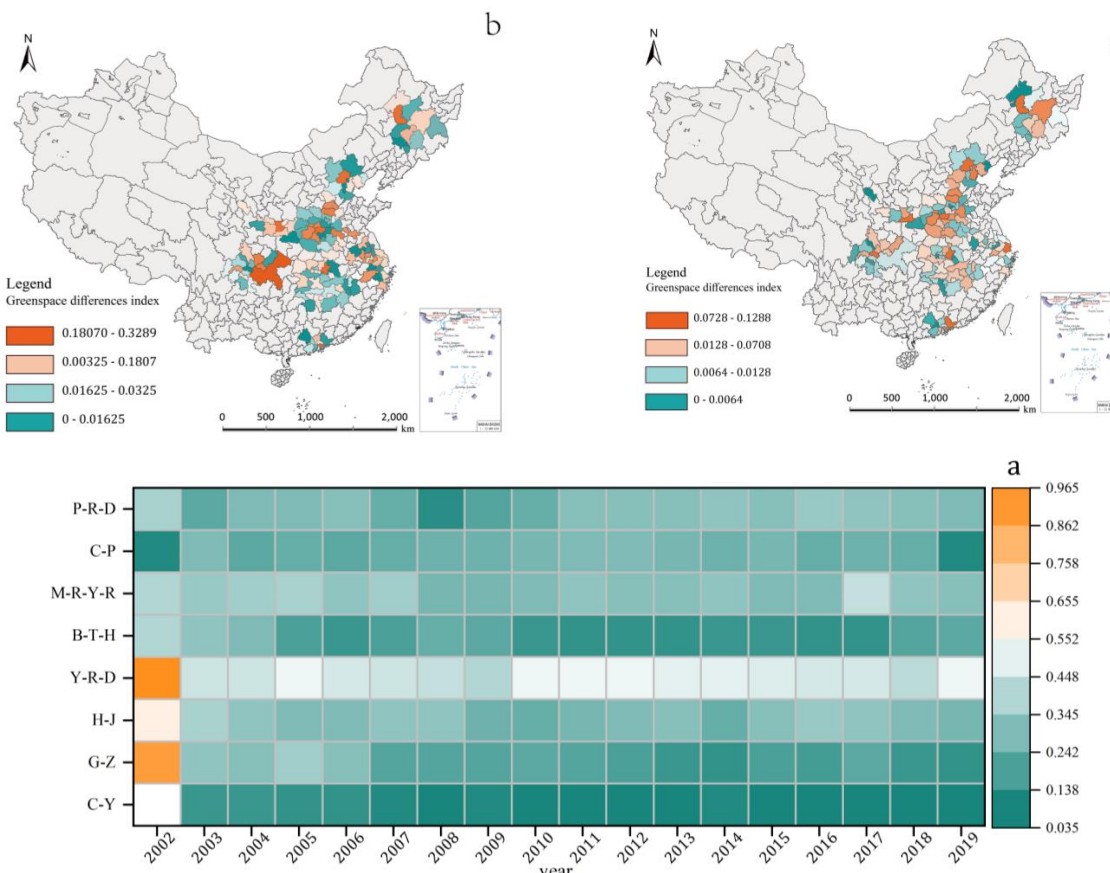

**Figure 7.** (**a**) The GEI (j) values of urban agglomeration from 2002 to 2019; (**b**) the GEI values of 133 cities in 2002; (**c**) the GEI values of 133 cities in 2019.

### 3.5. Driving Factors for the Distribution of GEIs

GeoDetector was used to conduct an environmental risk factor analysis of GEI (Y) for eight urban agglomerations. A *t* test with a significance level of 0.05 was used. The environmental risk factors and their proxy variables (X) included variables such as elevation, slope, slope direction, population density, GDP, average annual temperature (A-ATEMP), average annual humidity (A-ARH), and average annual precipitation (A-AP). Table 4 shows the results of the calculation of q values for all risk factors, and the results indicate

that the natural climate is significantly different from the other variables, with the average annual temperature having the highest q value, followed by the average annual humidity and average annual precipitation, indicating that the natural climate is the most dominant environmental factor among these variables in determining the spatial pattern of GEI. Geographic factors, such as slope and elevation, also influence the spatial distribution of the GEI. The low q value of socioeconomic factors indicates that they are not the main influencing factors of the GEI.

**Table 4.** Results of the spatial correlation analysis of eight urban agglomerations.

|  | Elevation | Terrain Slope | Aspect | GDP | Population Density | A-AP | A-ARH | A-ATEMP |
|---|---|---|---|---|---|---|---|---|
| q statistic | 0.05126 | 0.38855 | 0.01623 | 0.04109 | 0.03170 | 0.80564 | 0.82211 | 0.83738 |
| p value | 0.000 | 0.000 | 0.09499 | 0.000 | 0.000 | 0.000 | 0.000 | 0.000 |

Note: The spatial correlation analysis of eight urban clusters was performed via the factor detector in GeoDetector, in which there were 8 groups of categorical quantities for a total of 64 and 8 groups of variables Y, totaling 27,288.

After determining the dominant factor among the natural factors in the distribution pattern of the GEI, we conducted regression analyses on the average annual temperature, average annual humidity, and average annual rainfall, replacing GDP with disposable income per capita and population density with population size. By correlating the statistics of all eight urban agglomerations, the results displayed in Figure 8 and Table 5 show that the UGS of urban agglomerations is correlated with socioeconomic and natural conditions. Excellent natural conditions are the basis of UGSs, and good economic conditions can provide strong support for the development of greening. Moreover, the economy and the development of UGSs support each other, and ecologically sound cities attract a larger workforce due to their livability. Figure 8 shows that built-up area ($R^2 = 0.87$, $p < 0.05$), population size ($R^2 = 0.86$, $p < 0.05$), and GDP per capita ($R^2 = 0.62$, $p < 0.05$) were significantly and positively correlated with UGS. The average annual precipitation ($R^2 = 0.21$, $p < 0.05$), average annual temperature ($R^2 = 0.21$, $p < 0.05$), and average annual humidity ($R^2 = 0.12$, $p < 0.05$) were significantly and positively correlated with UGS. Specifically, socioeconomic factors exerted a more significant impact on UGS than natural factors. If urban agglomerations experience only a future increase in green rates rather than a reasonable allocation of UGSs based on population size, the GEI will worsen.

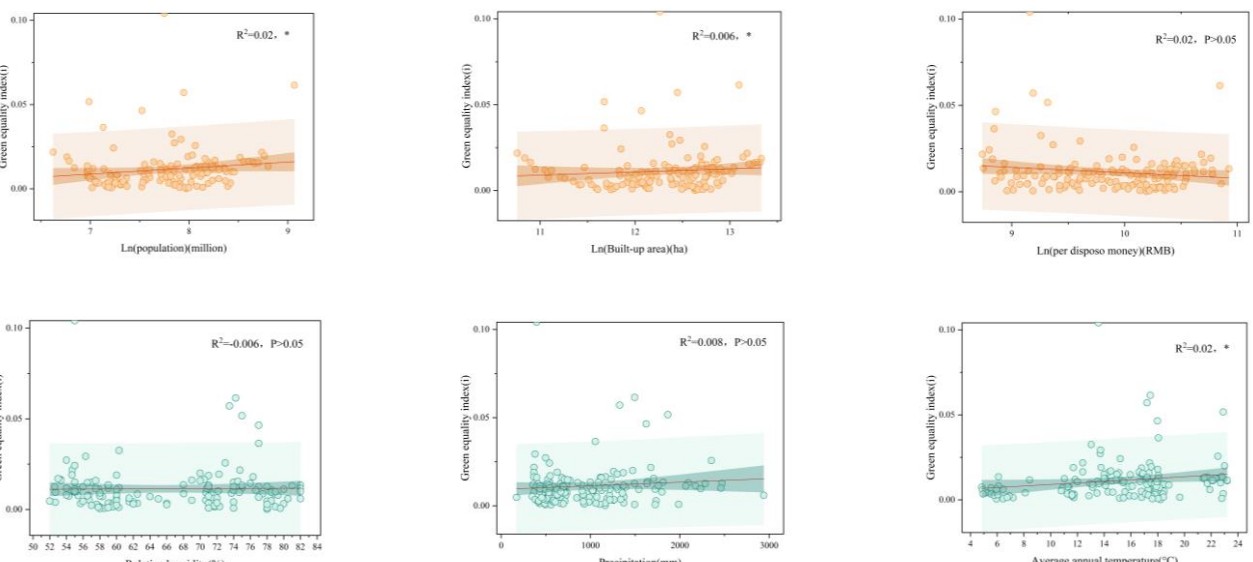

**Figure 8.** A linear fit using two dependent variables GEI (i), three socioeconomic factors, and three natural factors based on using the hydrothermal gradient values as independent variables. The thick orange and green lines in the figure represent the regression statistics corresponding to each independent variable, and $n = 144$. Significance levels were set as * $p \leq 0.05$.

**Table 5.** Spearman correlation results of potential factors among urban agglomerations.

|  |  | GEI (i) | BUA | PCGDP | A-ATEMP | A-ARH | A-AP | UP |
|---|---|---|---|---|---|---|---|---|
| **Green equality index (i)** | Spearman's r | 1 | 0.239 | −0.012 | 0.190 | −0.026 | 0.120 | 0.259 |
|  | *p* Value | – | 0.004 | 0.883 | 0.023 | 0.758 | 0.156 | 0.002 |
| **BUA** | Spearman's r | 0.239 * | 1 | 0.715 | 0.548 | 0.372 | 0.596 | 0.962 |
|  | *p* Value | 0.004 | – | 0 | 0 | 0 | 0 | 0 |
| **PCGDP** | Spearman's r | −0.012 | 0.715 | 1 | 0.150 | 0.140 | 0.217 | 0.669 |
|  | *p* Value | 0.883 | 0 | – | 0.072 | 0.093 | 0.010 | 0 |
| **A-ATEMP** | Spearman's r | 0.190 * | 0.548 | 0.150 | 1 | 0.699 | 0.841 | 0.537 |
|  | *p* Value | 0.023 | 0 | 0.072 | – | 0 | 0 | 0 |
| **A-ARH** | Spearman's r | −0.026 | 0.372 | 0.140 | 0.699 | 1 | 0.786 | 0.329 |
|  | *p* Value | 0.758 | 0 | 0.093 | 0 | – | 0 | 0 |
| **A-AP** | Spearman's r | 0.120 | 0.596 | 0.217 | 0.841 | 0.786 | 1 | 0.555 |
|  | *p* Value | 0.156 | 0 | 0.010 | 0 | 0 | – | 0 |
| **UP** | Spearman's r | 0.259 * | 0.962 | 0.669 | 0.537 | 0.329 | 0.555 | 1 |
|  | *p* Value | 0.002 | 0 | 0 | 0 | 0 | 0 | – |

Note: A two-tailed significance test was used, with one dependent variable GEI (i) and three socioeconomic factors and three natural factors used as independent variables. * Indicates a significant correlation at the 0.05 level.

## 4. Green Space Equity Optimization Strategy for Urban Agglomerations

### 4.1. Green Space Optimization Strategies for Urban Agglomerations under the Agglomeration Effect

The green development efficiency of urban agglomerations has spatial effects, which are usually manifested in the mutual influence between neighboring cities [13]. Studies related to the efficiency of green development in urban agglomerations have provided guidance for the allocation of green space resources in urban agglomerations from several perspectives [27,28]. Urban agglomerations need to set different green space construction goals according to their own specific development goals, population size, economic conditions, and geographical and natural bases [29]. Considering the disparity in economic status, urban agglomerations need to develop realistic improvement policies according to their socioeconomic conditions.

### 4.2. Strategies for Optimizing Green Space in Urban Agglomerations by Geographical and Natural Factors

Previous studies have long-established that climate factors have the strongest explanatory power for UGS differences in most geographic regions [30]. First, climate primarily controls the distribution patterns of vegetation [31]. Second, precipitation and temperature, as fundamental elements of climate zonation, largely influence the vegetation type and growth rate, which can have an impact on the overall amount of UGS [30]. This is consistent with the view advanced in this paper that physical geographic conditions are the main factors influencing the differences in green space distribution in urban agglomerations. We display the climatic zones to which urban agglomerations belong in Figure 9 to better contrast the influence of geoclimatic factors on the distribution of UGSs. We found that some urban agglomerations with excellent geoclimatic conditions, such as the P-R-D and Y-R-D urban agglomerations, still lack UGSs and show inequitable distributions of green space (Figure 10). This is due to the high population pressure in these urban agglomerations. Whether regarding either national or regional urban agglomeration, the scope and population of Chinese urban agglomerations are larger than those of other countries [11]. The large population base has completely diluted prior favorable climatic conditions [30].

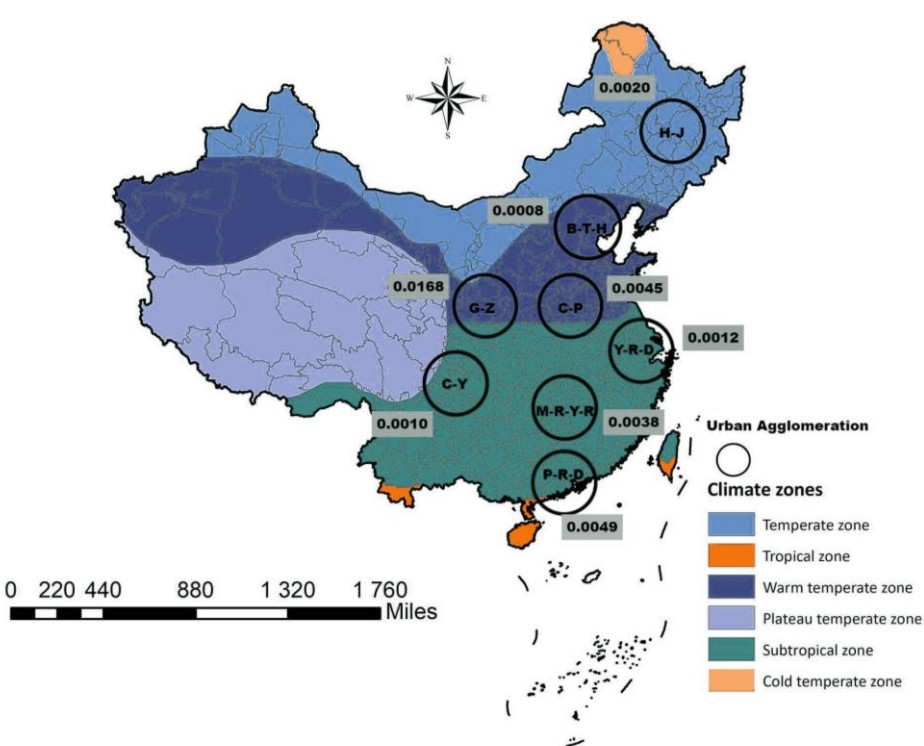

**Figure 9.** Relationship between the GEI (gray labels with numbers) of urban agglomerations and the climate zones to which they belong, using a climate zone map taken from [32].

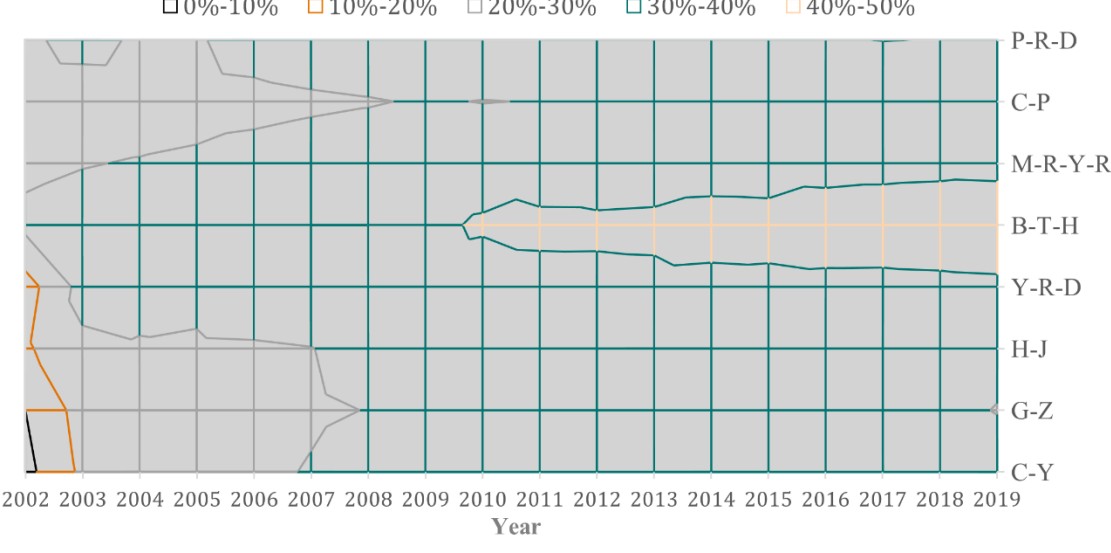

**Figure 10.** Changes in the ratio of built-up green area to built-up area from 2002 to 2019.

*4.3. Assessment of Green Space Resources and Policy Guidance*

To optimize the allocation of green space resources in urban agglomerations to achieve the best effect of land use within the region, policy makers first need to conduct a comprehensive survey and assessment of the existing green space resources to understand the quantity, types, distribution patterns, and changing trends of green space resources within the region. Sound green space planning objectives need to be established to reduce regional competition for green space resources in urban agglomerations [27]. Assessment should be conducted with the help of the various tools and techniques used in this paper, such as RS and GIS, as well as data analysis tools, such as the GEI index, Durbin model, and Gini coefficient. Second, the priorities related to green space resources need to be identified, the

types of green space in need of expansion need to be judged, the types of land use need to be regulated, and a regional green space resource allocation plan needs to be developed.

An increase in the amount of green space resources in urban agglomerations is achieved by promoting multisectoral government collaboration, encouraging public participation, optimizing environmental regulations, and enhancing the maintenance and management of green space resources [10]. For western urban agglomerations, noncentral cities, and developmental urban agglomerations, however, the potential scope of urban expansion is limited. Therefore, maintaining the existing amount of green space and preventing the compression and fragmentation of ecological land are the main priorities for improving the efficiency of green space utilization. The degree of urban green space equity is weaker in central cities, eastern urban agglomerations, and mature urban agglomerations; therefore, increasing the amount of green space in urban agglomerations, strengthening the level of policy guidance for urban agglomerations, and continuously optimizing resource allocation to ensure a positive cycle of ecological environment and economic development in urban agglomerations is necessary. The exchange and cooperation among city clusters and cities should also be strengthened to realize the secondary effective distribution of green space resources across regions.

## 5. Discussion

### 5.1. Green Space Distribution Pattern of Urban Agglomerations

The distribution and quantity of UGSs have changed greatly [21]. China, Europe, South America, and Latin America have different degrees of UGS distribution differences [33–36]. Especially in China, due to its vast territory, differences in the distribution of UGSs have been reported at the city scale [21], provincial scale [34,37], urban agglomeration scale [14], and even at regional scales [38]. Most studies have claimed that the quantity and quality of urban green space have improved significantly over the past 20 years. In contrast to Ze Xu's claim that the UGS difference outwardly weakens from the border between the B-T-H and Y-R-D, our results show that the distribution pattern of UGS differences is more inclined to inland–coastal and to economically developed-economically lagging forms of distribution. Similarly, Zhao believes that the increase in green space coverage follows a similar trend in large areas [21]. This paper confirms not only that differences in green space distribution exist at both the urban agglomeration scale and at the city scale but also that the disparities are more pronounced at the city scale.

### 5.2. The Development Process of UGS Equity in Different Cities

Specifically, during these 20 years, a higher level of UGS equity emerged in cities. We categorized these cities according to their population size. Among them, the first category included cities with obvious differences and large populations (Chongqing, Shanghai, Beijing, Tianjin, Shenzhen, Chengdu, Xi'an, Wuhan, Huizhou, Zhengzhou, Harbin, and Changchun); the second category included cities with smaller differences and large populations (Guangzhou, Zhuhai, Dongguan, Changsha, and Lanzhou); the third category included cities with obvious differences and small populations (Xianyang, Huzhou, Taizhou, Zhenjiang, Guangan, Deyang, and Daqing); and the fourth category included cities with smaller differences and small populations (all remaining cities).

The mega and large cities in Category 1 and Category 2 have a large number of UGSs, but because of their substantial populations, the weighted treatment yields a very low GEI. For Category 1, it is necessary to pay more attention to the equity of green space, increase the overall area of green space, and consider the distribution balance within the region. Most of the cities in Category 2 have good climatic conditions. They are warm and humid, which is ideal for plant growth and makes for excellent urban greenery.

Most of the cities in Category 3 are located in highly urbanized areas. For Categories 3 and 4, it is necessary to maintain the amount of ecological land in the process of urbanization and increase the quality of UGSs. Due to their close proximity to core cities, these cities tend to absorb an excessive amount of population; combined with limited green space,

these factors have led to their current classification. Category 4 represents the state of most cities in China, where small and medium-sized cities gradually come to consider greening to meet the greening standards set by the state and to eventually reach the average level of greening.

*5.3. Highly Urbanized Areas Have More Advanced Green Space Well-Being*

During the process of urban expansion, the land use pattern usually changes greatly [39]. The development and expansion of core cities consumes large amounts of ecological land [40], resulting in the degradation of their original nature [41]. In contrast to the fixed land use type in old urban areas, newly developed areas usually have more space to devote to new green space [42]. In this regard, we calculated the ratio of urban green space within the built-up area from 2002–2019 (Figure 10) and found that the proportion of green area/built-up area in Chinese urban agglomerations has remained at approximately 30–40% since 2006, and the data indicate strong convergence. On the other hand, this ratio is as high as 46% for the highly urbanized areas represented by the B-T-H, Y-R-D, and M-R-Y-R, which indicates that urbanization and green space well-being have a certain synergistic effect. Second, socioeconomic status also affects the degree of regional green space differences [43,44]. The primary factor needed for production and the driving force behind labor is population, which is a spatially heterogeneous factor for UGSs [45]. From an economic perspective, the economy initially exerts a certain positive effect on the supply of UGSs [37]. It is widely accepted that in the long run, urbanization generally improves the socioeconomic status of a region while reducing its provisioning of green space [42]. In this paper, we argue that socioeconomic conditions do not hinder the growth of green areas in the later stages of urban development but rather slow the growth rate of those areas. We found that the growth of the GEI gradually stabilizes among urban agglomerations, and the relevance of this conclusion can be interpreted from Figures 4 and 6.

*5.4. Multiple Influencing Factors That Affect the Distribution of Green Spaces in Urban Agglomerations*

The issue of regional disparities has been of major concern to geographers, economists, and government administrators. It is particularly important to note that geographical and historical factors may restrict regions with different geographical conditions from forming their own green space growth patterns in the early stage; that is, cities in the same region show basically the same rate of green space growth, but the level of those differences among different regions is still very large. This view aligns with the conclusions of this paper stating that natural and geographic conditions dominate the emergence of patterns of variation in green space distribution. Many studies have concluded that differences in economic status affect the equity of green space allocation. GDP per capita has a positive effect on both the urban green space growth rate and public green space per capita, while the urbanization rate, secondary industry, urban land, and population density have opposite effects on each of these two green space indicators [46,47]. In this study, we found that within urban agglomerations, the more urbanized a city is, the less equitable its green space, which is consistent with previous findings.

Urban agglomerations are central locations for capital, labor, and information [11]. In recent years, the Chinese government has approved an increasing number of official plans for the development of urban agglomerations. Since the birth of the first national-level urban agglomeration in 2010, six new national-level urban agglomerations have been formed as of the end of 2018. Additionally, policy regulation affects the growth of regional green space [37]. The "National Ecological Garden City" wave that was launched in 2004 was the first milestone in promoting the growth of UGSs. During this period, with the continuous advancement of national policies, urban governance regulations have gradually improved. The concept of improving the living environment has also been reflected in subsequent policies. The State Council officially promulgated the "National Garden City Selection Criteria" in 2016. The greening level of urban agglomerations in China has

improved. Through technological improvements and effective management [38], some areas that were originally less green have shown a good growth trend in recent years. Through the regulation of governmental management policies, some areas with interior geographical conditions can also achieve a balanced growth of green space [37,38].

*5.5. Limitations and Future Work*

Similar to the problems faced by others using economic difference indices to measure UGS, the GEI established in this paper, while capable of roughly determining the supply of UGS in a given urban agglomeration or city, is unable to demonstrate to policy makers the exact location of UGS disparities. Proper consideration of the impact of seasonal data on UGS distribution is also an issue that needs attention in the future. China's urbanization pace is expected to slow down over the next decade [48], a trend that provides an opportunity for urban agglomerations to focus on environmental sustainability. Thus, best practices in monitoring the changes in the urban ecological environment throughout the process of urbanization and maintaining the well-being of residents are topics that we will continue to explore in the future.

## 6. Conclusions

In this study, the GEI index was used to address the fundamental question of whether China's urban agglomerations undergo inequities in UGS distribution and management throughout the urbanization process. We also analyzed the relative contributions of spatial influencing factors and the socioeconomic determinants of this change. Based on the panel data of eight urban agglomerations (133 cities) at the local level and above, covering the period of 2002–2019, clear evidence is presented. Benefiting from the increase in UGSs in large regions (a net increase of 3.5 times), the equity in UGS distribution in major urban agglomerations in China has improved significantly, increasing from 3.74 in 2002 to 6.34 in 2019. As a substantive constituent of urban agglomerations, the level of UGS equality in Chinese cities has also improved. Of the 133 Chinese cities included in the study, 35% showed reduced variance in UGS distribution, with the median value of GEI decreasing from 0.0325 in 2002 to 0.0128 in 2019. According to the official green space index used to measure the differences in the distribution of UGSs in urban agglomerations and to identify the areas and cities with lower UGS equity, it is beneficial for decision makers to formulate more targeted greening policies and effectively improve green equity in urbanized areas.

This study also found that natural factors play a dominant role in the spatial pattern of the GEI, and the average annual temperature of a region shows a significant positive correlation with the GEI values of that region. In addition, GDP exerts an impact on the distribution of green space in urban agglomerations, which implies that the spatially polarized development of the economy could exacerbate the differences in the distribution of UGS in the future. Research on urban green space equity can help governments and urban planners better understand the current status of the ecological system and green space coverage across different areas of urban clusters, and thus, formulate more targeted and sustainable environmental protection and ecologically balanced plans.

**Author Contributions:** Conceptualization, X.Z. (Xueyan Zheng); Methodology, X.Z. (Xueyan Zheng); Software, X.Z. (Xueyan Zheng) and H.P.; Formal analysis, X.Z. (Xueyan Zheng); Investigation, X.Z. (Xueyan Zheng); Data curation, X.Z. (Xueyan Zheng), M.Z., H.P., W.N. and X.Z. (Xinyi Zhu); Writing—original draft, X.Z. (Xueyan Zheng), M.Z. and X.N.; Writing—review & editing, X.Z. (Xueyan Zheng) and Y.S.; Supervision, Y.S., W.N., G.Y. and Z.B.; Funding acquisition, Y.S. All authors have read and agreed to the published version of the manuscript.

**Funding:** This research was financially supported Zhejiang Provincial Natural Science Foundation of China under Grants No. LY19C160007.

**Institutional Review Board Statement:** Not Applicable.

**Informed Consent Statement:** Not Applicable.

**Data Availability Statement:** Not Applicable.

**Conflicts of Interest:** The authors declare no conflict of interest.

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
