# Peer review of "Equity Analysis of the Green Space Allocation in China’s Eight Urban Agglomerations Based on the Theil Index and GeoDetector"

_land, doi:10.3390/land12040795_

Round 1

Reviewer 1 Report

Equity Analysis of the Greenspace Allocation in China's Eight Urban Agglomerations Based on the Theil Index and GeoDetector

 This paper evaluates urban greenspace spatial distribution of eight urban agglomerations in China in nearly twenty years. A new index is defined based on Theil index. Geodetector is used to measure spatial correlation of urban greenspaces.

 Following suggestions can be considered to further improve the quality of this article:

1.      English language editing.

2.      In four categories of urban agglomerations mentioned in the second section, how to distinguish the two “regional”?

3.      Please further verify “…the total built-up area of the study area is 2,748,327 ha, including an urban population of 28,670 million.”. Also the population listed in table 4 need further verification.

4.      What is UGS? And some other abbreviations need explanations.

5.      The boundary definition influences further greenspace analysis in urban areas. It should be accurately defined. It is not clear stated in the description “only analyzes the areas with urban attributes…….The built-up area refers to the area with well-developed urban construction”.

6.      Also it is not clearly listed which kind of socio-economic data is in section 2.2.1.

7.      How does this work handle the inconsistency of resolutions of different data?

8.      What is the seasons of data acquisition of urban greenspace for different agglomeration?

9.      The relationship between Theil index and GEI is not clearly described in functions (1), (2) and (3).

10.   It is necessary to explain the roles of GEI and Geodetector correspondingly in this work.

Author Response

Reviewer #1 comments

Equity Analysis of the Greenspace Allocation in China's Eight Urban Agglomerations Based on the Theil Index and GeoDetector. This paper evaluates urban greenspace spatial distribution of eight urban agglomerations in China in nearly twenty years. A new index is defined based on Theil index. Geodetector is used to measure spatial correlation of urban greenspaces.

Following suggestions can be considered to further improve the quality of this article:

Response: Thank the reviewer for these comments. We are grateful for your recognition of our research. Based on the comments listed below, we have carefully revised the entire manuscript.

  1. English language editing.

Response: Thanks for the constructive suggestions to improve our manuscript. We use AJE's premium touch-up service. The language of the manuscript was revised by the professional language editing company.

  1. In four categories of urban agglomerations mentioned in the second section, how to distinguish the two “regional”?

Response: Thanks for your grateful indication. We delete the second ‘regional’ as a clerical error. In addition, we have simplified 2.1 to more clearly describe our study area. (Page3, Lines 99-124)

  1. Please further verify “…the total built-up area of the study area is 2,748,327 ha, including an urban population of 28,670 million.”. Also the population listed in table 4 need further verification.

Response: Thank you for this suggestion which is very much appreciated. The typo was amended. We modified the original ambiguous sentence to ‘By the end of 2019, the built-up area covered by this study accounted for 45.6 % of the country, and the radiation population reached 65.9 % of the national urban population.’  (Page3, Lines109-110)

  1. What is UGS? And some other abbreviations need explanations.

Response: We thank the reviewers for their careful reading of the manuscript. We have added an explanation of abbreviations in 2.2 and added the source of the definition of abbreviations in the footnotes of Table 1. The new additions to the literature are as follows: (Page4, Lines126; Page5, Lines151-161)

National Bureau of Statistics of China. China City Statistical Yearbook. Beijing: China Statistics Press; 1990–2010 [Chinese].

  1. The boundary definition influences further greenspace analysis in urban areas. It should be accurately defined. It is not clear stated in the description “only analyzes the areas with urban attributes…….The built-up area refers to the area with well-developed urban construction”.

Response: We are very grateful for the reviewer's comments. We refer to the definition of built-up areas by the Ministry of Housing and Urban-Rural Development and published academic studies, and redescribe the definition of built-up areas. (Page5, Lines151-154)

  1. Also it is not clearly listed which kind of socio-economic data is in section 2.2.1.
    Response: Thanks for the reviewer's careful review. The type of socio-economic data has been increased to 2.2 (Page5, Lines162-163)
  2. How does this work handle the inconsistency of resolutions of different data?
    Response: According to the reviewer’s suggestions, we have unify the resolution of the raster data. (Page5, Lines163-172)
  3. What is the seasons of data acquisition of urban greenspace for different agglomeration?

Response: Thank you for your comments that have focused our attention on possible future research directions. Our study did not include seasonal data, but this is an interesting perspective and we will consider this proposal in future studies. Therefore, we put this point in 5.5. (Page19, Lines566-567)

  1.  The relationship between Theil index and GEI is not clearly described in functions (1), (2) and (3).
    Response: Thank you for your feedback, it was much appreciated. We did refine the Theil index to measure differences in urban clusters, and they are related as follows.

According to the Theil index:

T=∑_(i=1)^Ni▒(xi/X) ×ln⁡{(xi/X)/(yi/Y)}

(1)

Then, the overall difference is:

GEI= Ni – [GEI (i)+GEI (j)]

The between-group component is:

GEI (i)= Σ (X i/X) ×ln [ (X i/X)/(Y i/Y)]          (2)

The within-group component is:

GEI (j)= Σ (x i/X i) ×ln [ (x i j/X i)/(y i j/Y i)]      (3)

We have enhanced the description of the formula part to show exactly the deformation relationship and transformation of Theil index and GEI index by mathematical formula. (Page7, Lines202-212).

  1. It is necessary to explain the roles of GEI and Geodetector correspondingly in this work.
    Response: Thank you for your careful review and constructive suggestions regarding our manuscript. According to your advice, we supplement an explanatory paragraph added to the flow chart in this 2.3. The role of GEI is expressed graphically. (Page6, Lines174-185).

Reviewer 2 Report

This study provides meaningful information on the analyzing a Greenspace Equity Index (GEI) of the China's eight urban agglomerations from 2002 to 2019, as well as to consider the possibility of using Theil index combined with GeoDetector to measures spatial correlation of greenspace resources. The study is useful and generally clearly presented. However, there are some fundamental error, and the presentation of data can be improved. My comments are below:

1)      The original article found many typos, and the manuscript should be proofread carefully.

2)      Motivation and literature section is slightly handled. Author needed to improve it substantially.

3)      Material and methods sections are hard to understand because the research steps are not adequately described. I would recommend the introduction of an explicit phased methodological diagram or technical flow diagram.

4)      The accuracy of the driving force analysis depends on raster data, the reason for the authors to select the raster 1 km × 1 km resolution for represents the total GDP output value of the urban agglomeration GDP data, in contrast with the average temperature, humidity and precipitation data were obtained from the 500 m × 500 m raster.

5)      Discussions need to be more scientific by highlighting more clearly how the analysis proposed by the authors can support the implementation of the measures proposed by the greenspace allocation in China's eight urban agglomerations.

Author Response

Reviewer #2 comments

This study provides meaningful information on the analyzing a Greenspace Equity Index (GEI) of the China's eight urban agglomerations from 2002 to 2019, as well as to consider the possibility of using Theil index combined with GeoDetector to measures spatial correlation of greenspace resources. The study is useful and generally clearly presented. However, there are some fundamental error, and the presentation of data can be improved. My comments are below:

Response: Dear reviewer, Thank you for reviewing our manuscript and for the constructive comments, which greatly helped us to improve the manuscript.

  1. The original article found many typos, and the manuscript should be proofread carefully.

Response: Thanks for the constructive suggestions to improve our manuscript. We use AJE's premium touch-up service. The language of the manuscript was revised by the professional language editing company.

  1. Motivation and literature section is slightly handled. Author needed to improve it substantially.
    Response: We appreciate and are grateful for your valuable suggestions. We add the contribution and significance of this study to the sustainability of current urban agglomerations.

We agree that the first two paragraphs of the introduction were weak and have rewritten them to be clearer. A deeper dive into the background and motivation of the study was made in the introduction section. (Page1-2, Lines36-87)

At the same time, we supplemented the progress of green space research in urban agglomerations with a new reference in the second paragraph of the introduction. (Page2, Lines49-59)

The value and practical significance of the study is discussed in the third paragraph of the introduction. (Page2, Lines68-72)

  1. Material and methods sections are hard to understand because the research steps are not adequately described. I would recommend the introduction of an explicit phased methodological diagram or technical flow diagram.

Response: Thanks for the reviewer's careful review. According to your advice, we supplement a technical flow diagram added to the flow chart in this 2.3. The role of GEI is expressed graphically. (Page6, Lines174-185)

Figure 2. Work flow chart for this study.

  1. The accuracy of the driving force analysis depends on raster data, the reason for the authors to select the raster 1 km × 1 km resolution for represents the total GDP output value of the urban agglomeration GDP data, in contrast with the average temperature, humidity and precipitation data were obtained from the 500 m × 500 m raster.

Response: We appreciate your comment, thank you for pointing it out. According to the reviewer’s suggestions, we have unify the resolution of the raster data. We standardized the resolution of the gridded data to 1km*1km. (Page5, Lines163-172)

  1.  Discussions need to be more scientific by highlighting more clearly how the analysis proposed by the authors can support the implementation of the measures proposed by the greenspace allocation in China's eight urban agglomerations.

Response: Thank you very much for your comments. We modified the discussion section and reconstructed the logical framework of the section. It is developed from the following aspects:

  • Green space distribution pattern of urban agglomerations
  • The development process of UGS equity in different cities
  • Highly urbanized areas have more advanced greenspace well-being
  • Multiple influencing factors that affect the distribution of greenspaces in urban agglomerations
  • Limitations and future work

(Page17-19, Lines463-561)

In addition, in order to develop optimization strategies for green space distribution in urban agglomerations in more detail, we have added part 4. (Page13-15, Lines402-461)

Reference

National Bureau of Statistics of China. China City Statistical Yearbook. Beijing: China Statistics Press; 1990–2010 [Chinese].

Zhao, J., Chen, S., Jiang, B., Ren, Y., Wang, H., Vause, J., Yu, H., 2013. Temporal trend of green space coverage in China and its relationship with urbanization over the last two decades. Sci. Total Environ. 442, 455-465.

Cao, L., Huo, X., Xiang, J., Lu, L., Liu, X., Song, X., Jia, C., Liu, Q., 2020. Interactions and marginal effects of meteorological factors on haemorrhagic fever with renal syndrome in different climate zones: Evidence from 254 cities of China. Sci. Total Environ. 721, 137564.

Harris, I., Osborn, T.J., Jones, P. et al. Version 4 of the CRU TS monthly high-resolution gridded multivariate climate dataset. Sci Data 7, 109 (2020). https://doi.org/10.1038/s41597-020-0453-3.

Reviewer 3 Report

This is a very interesting and well constructed /written paper. Presented problems are very current and important. Article concerns sustainable development of cities, case studies: 8 China agglomerations, in the range of the correlation between built-ap area and Urban Greenspaces (UGS) area. It is extremly current problem concerning "cities of tomorrow": how to plan develolopment of cities to keep compact built-up area and, at the same time, increase the Greespaces areas as a dense network, in which the building is immersed.

The Authors present extensive research results, rich literature analysis and variety of factors contributing changes in Green Areas share in relation to built-up area changes, including distribution of these changes. This is a very interesting / valuable material which increases the knowledge in city planning and reveal some mechanisms / regularities occuring in city development. Research has been methodicaly well planned.

Detailed suggestions:

Line 88 - rather: "...and local..." instead of existing "regional" (twice "regional" is)

Table 2 - there is a little mess in this table - in the column "Dependent variables". The suggestion is to put "climatic parametrs" (average annual temperature, average annual humidity and average annual precipitation) next to each other - after the rest of parameters.

Line 508 - rather "interior" - not "inferior".

In general, I estimate the paper highly as the most important - giving materials / guidelines to modern city planning and development leading / bringing us closer to green cities - cities of future. This paper is worth publishing after minor revisions.

Author Response

Reviewer #3 comments

This is a very interesting and well constructed /written paper. Presented problems are very current and important. Article concerns sustainable development of cities, case studies: 8 China agglomerations, in the range of the correlation between built-ap area and Urban Greenspaces (UGS) area. It is extremly current problem concerning "cities of tomorrow": how to plan develolopment of cities to keep compact built-up area and, at the same time, increase the Greespaces areas as a dense network, in which the building is immersed.

The Authors present extensive research results, rich literature analysis and variety of factors contributing changes in Green Areas share in relation to built-up area changes, including distribution of these changes. This is a very interesting / valuable material which increases the knowledge in city planning and reveal some mechanisms / regularities occuring in city development. Research has been methodicaly well planned.

Detailed suggestions:

Response: Dear reviewer, thanks for your careful reading of the manuscript and their constructive remarks. We have taken the comments on board to improve and clarify the manuscript. Please find below a detailed point-by-point response to all comments . Sincerely, Xueyan Zheng.

  1. Line 88 - rather: "...and local..." instead of existing "regional" (twice "regional" is)

Response: Thanks for your reminder. We delete the second ‘regional’ as a clerical error. In addition, we have simplified 2.1 to more clearly describe our study area. (Page3, Lines 99-124)

  1. Table 2 - there is a little mess in this table - in the column "Dependent variables". The suggestion is to put "climatic parametrs" (average annual temperature, average annual humidity and average annual precipitation) next to each other - after the rest of parameters.

Response: Thanks for the reviewer's careful review. We have placed the "climate parameters" (average annual temperature, average annual humidity and average annual precipitation) after the other parameters. (Page6, Lines 173-174)

  1. Line 508 - rather "interior" - not "inferior".

Response: Thanks for the constructive suggestions to improve our manuscript. And sorry for the language problem, we had improved the language using a language editing service. The language of the manuscript was revised by the professional language editing company.

I would like to thank the reviewers and editor again for their critical comments, which are very helpful for the revision of this manuscript and also for our subsequent research. In addition to our point-by-point replies to the reviewers’ comments above, we also did our best to meet the standards of required editorial corrections and have made all changes easily identifiable. We hope that our revised manuscript meets your requirements. If any further action is needed, please let us know immediately. We look forward to hearing back from you.

Reference

National Bureau of Statistics of China. China City Statistical Yearbook. Beijing: China Statistics Press; 1990–2010 [Chinese].

Zhao, J., Chen, S., Jiang, B., Ren, Y., Wang, H., Vause, J., Yu, H., 2013. Temporal trend of green space coverage in China and its relationship with urbanization over the last two decades. Sci. Total Environ. 442, 455-465.

Cao, L., Huo, X., Xiang, J., Lu, L., Liu, X., Song, X., Jia, C., Liu, Q., 2020. Interactions and marginal effects of meteorological factors on haemorrhagic fever with renal syndrome in different climate zones: Evidence from 254 cities of China. Sci. Total Environ. 721, 137564.

Harris, I., Osborn, T.J., Jones, P. et al. Version 4 of the CRU TS monthly high-resolution gridded multivariate climate dataset. Sci Data 7, 109 (2020). https://doi.org/10.1038/s41597-020-0453-3.

Round 2

Reviewer 1 Report

Generally the revisions addressed many questions previously raised.

Still please do a language editing, for example "the radiation population" may not be appropriate considering the context.

Author Response

Response to the Reviewers’ Comments on Land-2272765

Dear Editor,

Thank you for your letter and for the reviewers’ comments concerning our manuscript entitled “Equity Analysis of the Greenspace Allocation in China's Eight Urban Agglomerations Based on the Theil Index and GeoDetector” (Land-2272765). These comments are all valuable and very helpful for revising and improving our paper, as well as the important guiding significance to our researches. We have considered the comments carefully and have made the following revisions which we hope meet with approval.

For the opinion of reviewer 1, we used AJE's advanced editing services to reduce language errors.The embellishment certificate is attached.Please see the attachment.
